# Association between Remnant Cholesterol and Metabolic Syndrome among Chinese Adults: Chinese Nutrition and Health Surveillance (2015–2017)

**DOI:** 10.3390/nu16193275

**Published:** 2024-09-27

**Authors:** Fusheng Li, Hongtao Yuan, Shuya Cai, Wei Piao, Jing Nan, Yuxiang Yang, Liyun Zhao, Dongmei Yu

**Affiliations:** 1National Institute for Nutrition and Health, Chinese Center for Disease Control and Prevention, Beijing 100050, China; 18246668833@163.com (F.L.); yuanht@ninh.chinacdc.cn (H.Y.); caisy@ninh.chinacdc.cn (S.C.); piaowei@ninh.chinacdc.cn (W.P.); nj13939012762@163.com (J.N.); yangyuxiang1996@sina.com (Y.Y.); zhaoly@ninh.chinacdc.cn (L.Z.); 2NHC Key Laboratory of Public Nutrition and Health, Chinese Center for Disease Control and Prevention, Beijing 100050, China

**Keywords:** remnant cholesterol, metabolic syndrome, biomarkers, prevalence

## Abstract

Background: Remnant cholesterol (RC) is highly associated with several chronic diseases. However, the relationship between RC and Metabolic syndrome (MetS) remains unclear. The study’s objective is to illustrate the relationship of RC to MetS. Methods: The data were collected from the Chinese Nutrition and Health Surveillance (2015–2017), which included personal, household and dietary information. A total of 65,618 residents aged 20 years or older from 31 provinces in mainland China were included in this study. RC was calculated by the equation RC = TC − (LDL-C + HDL-C). The criteria for MetS were based on the 2020 Chinese Type 2 Diabetes Prevention and Treatment Guidelines. Logistic regression models were used to analyse the relationship between RC and MetS and every MetS component. The receiver operating characteristic (ROC) curve was established to evaluate the accuracy of RC in identifying MetS, and the area under the curve (AUC) and the best threshold were calculated. Results: The weighted RC level of Chinese residents aged 20 years or older was 0.48 mmol/L. Participants with high RC levels were likely to be elderly, have a higher prevalence of MetS, higher total cholesterol (TC), triglyceride (TG), low-density lipoprotein cholesterol (LDL-C), fasting blood glucose (FBG), hba1c, and lower high-density lipoprotein cholesterol (HDL-C). Sex, body mass index (BMI), education status, household yearly income per capita, marital status, area of the country, residence location, smoking status, fruit intake and sleep time had statistical differences in the RC group (*p* < 0.05). The OR of MetS gradually increased with an increase in the RC quartile (*p* < 0.01), and higher quartiles of RC (Q4) suggested the highest MetS risk. The prevalence of each MetS component gradually increased with an increase in the RC quartile. The ROC curve found that to identify MetS, the AUC and best threshold of RC were 0.71 and 0.52 mmol/L, respectively. Conclusions: RC had a positive association with MetS and each MetS component. The accuracy in identifying MetS was higher in RC than in other indexes. The current study could provide new scientific evidence for the early prevention and control of MetS.

## 1. Introduction

Metabolic syndrome (MetS) is a disease characterised by obesity, hyperglycemia, dyslipidemia, and hypertension, which can cause serious harm to health [1]. As society continues to develop, the prevalence of various common chronic diseases is on the rise. The global prevalence of MetS was estimated to be around 25%, and according to the Chinese Health and Nutrition Survey, the prevalence of MetS among Chinese residents had increased from 21.3% in 2009 to 31.1% in 2015–2017, which was much higher than the global average [2,3]. Relevant studies have shown that MetS was highly associated with the occurrence of cardiovascular disease. A Korean cross-sectional study found that MetS was positively associated with the severity of CVD and was associated with abnormal changes in the left ventricular structure and diastolic function [4]. Moreover, MetS was highly associated with sleep apnea syndrome, chronic kidney disease, peripheral neuropathy, and other diseases, causing significant threats to health [5,6,7].

Remnant cholesterol (RC) is receiving increasing attention from scholars around the world as one of the potential causes of chronic diseases, including atherosclerotic cardiovascular disease, hypertension, and non-alcoholic fatty liver [8,9]. RC refers to the remaining part of total cholesterol that is separated from HDL cholesterol and LDL cholesterol and is carried within very-low-density lipoproteins (VLDL), intermediate-density lipoproteins (IDL), and chylomicron remnants [10]. RC is capable of directly penetrating arterial walls, which are engulfed by vascular macrophages and smooth muscle cells [11]. Then, the endothelial surface cells transition to foamy macrophages, developing an unstable complex plaque and promoting the occurrence and development of atherosclerotic cardiovascular diseases [11]. Previous studies have found a positive correlation between RC and chronic diseases such as hypertension, obesity, dyslipidemia, and diabetes [12,13,14,15], all of which are components of MetS and contribute to its development. However, the current relationship between RC and MetS still needs to be clarified. This study used the latest nationally representative data of China to investigate the relationship between RC and MetS, as well as to evaluate the accuracy of RC in diagnosing MetS. The results of this study will provide new scientific evidence for the early prevention of MetS.

## 2. Materials and Methods

### 2.1. Data Source

Data were obtained from the Chinese Nutrition and Health Surveillance (2015–2017) [16], a nationally representative survey in mainland China. The residents living in 31 provinces at 298 survey sites were surveyed in 2015. The survey adopted a stratified, multistage, random sampling method. Household questionnaire surveys and dietary evaluations were used to investigate the participants. This study was approved by the Ethics Committee of the National Institute for Nutrition and Health and the Chinese Center for Disease Control and Prevention. The ethical approval numbers were 201519-A and 201614. All participants provided informed consent before the study. A total of 65,618 participants aged 20 years or older were included in the analysis.

### 2.2. Surveillance Contents

The Chinese Nutrition and Health Surveillance (2015–2017) collected data in four parts: a questionnaire survey, physical examination, laboratory tests, and a dietary survey. Individual information, body measurements, biochemical indicators and food consumption were included in this study.

#### 2.2.1. Questionnaire Survey

A face-to-face questionnaire was used to collect household and personal information. The household questionnaires mainly collected basic information about the household members, household economic status and other information. Personal questionnaires mainly collected the prevalence, control and family history of chronic diseases, as well as smoking, drinking, dietary habits and physical activity status.

#### 2.2.2. Physical Examination

Height was measured using a metal column height gauge, which was accurate to 0.1 cm. Weight was measured using an electronic weight scale, which was accurate to 0.1 kg. Blood pressure was measured using an electronic sphygmomanometer, which was accurate to 0.1 mmHg. All measuring instruments met the national metrological certification requirements.

#### 2.2.3. Laboratory Tests

An overnight fasting blood sample was collected from each participant to measure blood biochemical indexes. Fasting blood glucose (FBG) was determined using the glucose oxidase method, while hba1c was measured using high-performance liquid chromatography. Total cholesterol (TC) was measured using the cholesterol oxidase amino antipyrine phenol method, and triglyceride (TG) was measured using the phosphoglycerol-oxidase 4-chloric acid method. High-density lipoprotein cholesterol (HDL-C) and low-density lipoprotein cholesterol (LDL-C) were measured using the direct method.

#### 2.2.4. Dietary Survey

The food frequency questionnaire (FFQ) was used to collect the food consumption frequency and habits of the participants over the past year. FFQ evaluated the nutritional status of participants based on the frequency or types of various foods consumed daily, weekly, monthly, or annually. This study used the daily consumption of the participants’ fruits, vegetables, and red meat.

### 2.3. Quality Control

To ensure quality, the National Project Working Group developed 4 unifiers: unified programmes, manuals and questionnaires; unified equipment and reagents; unified training and assessment; and unified data entry and cleaning.

### 2.4. Definition of Variables

#### 2.4.1. Remnant Cholesterol

RC includes the cholesterol carried by VLDL, IDL and its residue, which can be estimated by the international common estimation method, RC = TC − (LDL-C + HDL-C), and this method has been recognised as the most common calculation method in the related research of primary and secondary prevention of RC [17].

#### 2.4.2. Metabolic Syndrome

The criteria of MetS was based on the 2020 Chinese Type 2 Diabetes Prevention and Treatment Guidelines, and participants meeting more than 3 of the following 5 criteria were defined as having the MetS:(1)High WC: WC ≥ 90 cm for male; WC ≥ 85 cm for female;(2)Elevated blood pressure: SBP  ≥  130 mm Hg or DBP  ≥  85 mm Hg or treatment of previously diagnosed hypertension.(3)High TG: serum TG ≥ 1.70 mmol/L or specific treatment for TG abnormality;(4)Low HDL-C: HDL-C ≤ 1.04 mmol/L or specific treatment for TG abnormality;(5)Elevated FBG: FBG ≥ 6.1 mmol/L or previously diagnosed type 2 DM.

#### 2.4.3. Covariates

A wide range of covariates were accounted for:(1) body mass index (BMI) was categorised as normal (<24 kg/m^2^), overweight (24≤ to <28 kg/m^2^) or obese (≥28 kg/m^2^); (2) education level was categorised as primary school graduate or below, middle/high school or college graduate or above; (3) according to the income quartile, income was categorised as no given, <10,000 CNY, 10,000–20,000 CNY or >20,000 CNY; (4) marital status was categorised as never married, married or other (cohabitation, widowhood, divorce, precinct); (5) area of the country was categorised as north or south; (6) residence location was categorised as urban or rural; (7) smoking was categorised as never smoked, formerly smoked or currently smoke; (8) alcohol consumption was categorised as never consumed, moderately consume (men consume less than 25 g of alcohol and women less than 15 g of alcohol per day) or excessively consume (men consume more than 25 g and women more than 15 g per day) [3]; (9) red meat intake was categorised as normal (daily average red meat intake ≤100 g) or excessive (>100 g) [18]; (10) vegetable intake was categorised as normal (daily average vegetable intake ≥400 g) or insufficient (<400 g) [18]; (11) fruit intake was categorised as normal (daily average fruit intake ≥300 g) or insufficient (<300 g) [18]; (12) sleep time was categorised as low (daily average <7 h), normal (7–9 h) or high (>9 h) [19]; (13) physical activity was defined as inactive (if, within one week, the total time of moderate-intensity activity was less than 150 min, or high-intensity activity was less than 75 min, or the cumulative amount of moderate- and high-intensity activity was less than 150 min), or active (if, within one week, the total time of moderate-intensity activity was more than 150 min, or high-intensity activity was more than 75 min, or the cumulative amount of moderate- and high-intensity activity was more than 150 min) [20].

### 2.5. Statistical Analysis

All analyses were conducted with SAS software (v.9.4, SAS Institute Inc., Cary, NC, USA), and the plot in this study was created using R software (Version 4.1.2). The categorical data were reported as numbers (percentages), and the Rao–Scott Chi-square test was performed to compare the distribution of different characteristics between subgroups. The continuous data were reported as $\bar{X}$ ± sd and the *t*-test was performed to compare the distribution of different characteristics between subgroups. In order to obtain national representativeness, the proc surveyfreq and proc surveymeans programs were applied. The weight of the sample was calculated by data from the Chinese National Bureau of Statistics in 2010.

The relationship between RC and MetS and every MetS component was analysed using a logistic regression model. Participants were divided into groups according to the RC level quartile: Group Q1 (RC ≤ 0.38 mmol/L), group Q2 (0.38 mmol/L < RC ≤ 0.46 mmol/L), group Q3 (0.46 mmol/L < RC ≤ 0.57 mmol/L), and group Q4 (RC > 0.57 mmol/L). We used Q1 of RC as the reference group to calculate the OR and 95% CI of other RC groups. We used a Spearman correlation plot to show the relationships between the MetS components and showed correlation coefficients. A compound bar chart was used to show the prevalence of MetS components in different RC groups, and a percentage bar chart was used to show the percentage of the number of MetS components in different RC groups.

We constructed a receiver operating characteristic (ROC) curve to evaluate the accuracy of RC in identifying MetS and calculated the area under the curve (AUC) and the best threshold. A two-sided *p*-value < 0.05 was considered to indicate statistical significance.

## 3. Results

### 3.1. General Characteristics of the Participants

A total of 65,618 participants were included in this study, including 30,363 males (46.3%) and 35,255 females (53.7%). Males had a higher level of RC than females (male: 0.50 mmol/L vs. female: 0.49 mmol/L, *p* < 0.05). Moreover, there were significant differences between males and females in red meat, vegetable, and fruit intake (*p* < 0.05). TG and FPG were significantly higher in men than in women, while MetS prevalence, TC, LDL-C, and HDL-C were significantly lower in men than in women (*p* < 0.05, Table 1).

### 3.2. Weighted General Characteristics of Adults Aged 20 and Older in 2015–2017

The weighted RC level of Chinese residents aged 20 years or older was 0.48 mmol/L. Participants with high RC levels were likely to have a higher age, higher prevalence of MetS, higher TC, TG, LDL-C, FBG, hba1c, and lower HDL-C. Sex, BMI, education status, household yearly income per capita, marital status, area of the country, residence location, smoking status, fruit intake, and sleep time had statistical differences in the RC group (*p* < 0.05, Table 2).

### 3.3. The Relationship between RC and MetS

To explore the relationship between RC and MetS, three models were established to adjust confounding factors. The results showed that from Model 1 to Model 3, compared with the lowest quartile group, RC and MetS were positively correlated, and the degree of correlation remained stable. Based on the results of the analysis of the correlation between RC quartile and MetS, the OR of MetS gradually increased with an increase in RC quartile (*p* < 0.01), and higher quartiles of RC (Q4) suggested the highest MetS risk. In Model 3, when people were in the Q4 group, the risk of MetS increased by 566% (Table 3).

RC and the components of MetS were positively associated. The results showed that from Model 1 to Model 3, compared with the lowest quartile group, RC and MetS components were positively correlated, and all models were statistically significant. With the increase in RC, the OR of each MetS component gradually increased. In the Q4 group, the highest OR between MetS components was Elevated TG (13.94), followed by Low HDL-C (3.57), Elevated FPG (2.05), High BP (1.51), High WC (1.47) (Table 4).

Association analysis showed that there were positive associations between RC and MetS components. The highest association between RC and MetS components was Elevated TG (0.4), followed by Low HDL-C (0.2) and other components (0.1) (Figure 1).

The prevalence distributions of High WC, High BP, Elevated TG, Low HDL-C and Elevated FPG among RC groups were statistically significant (*p* < 0.001). The prevalence of High WC, High BP, Elevated TG, Low HDL-C and Elevated FPG was gradually increased with an increase in the RC quartile. The prevalence of each component was the highest in the Q4 group. In the Q4 group, the highest prevalence of MetS components was Elevated TG (61.9%), followed by High BP (60.3%), Low HDL-C (45.3%), High WC (40.7%), and Elevated FPG (17.2%) (Figure 2).

RC and the number of MetS components had an association, with statistically significant differences in the number of MetS components among different RC groups (*p* < 0.001). With increasing RC, the proportion of two, three, four and five components increased, and the proportion of 0 and 1 components decreased. In the Q4 group, the proportion of 2 components was the largest, followed by 3, 1, 4, 0, and 5 components (Figure 3).

### 3.4. Accuracy of RC for Identifying MetS

ROC curve was established to evaluate the accuracy of RC in identifying MetS. As shown in Figure 4, RC showed normal diagnostic accuracy in identifying MetS, and the AUC and best threshold in the population were 0.71 and 0.52 mmol/L, respectively. The accuracy in identifying MetS showed significant differences by sex (*p* < 0.01), and the AUC and best threshold in males and females were 0.70, 0.73 and 0.54 mmol/L, 0.51 mmol/L, respectively. The sensitivity in identifying MetS was low (0.60), but the specificity in identifying MetS was high (0.72). Different sexes also affected the sensitivity and specificity in identifying MetS. The sensitivity and specificity in males and females were 0.59, 0.65 and 0.71, 0.70, respectively.

## 4. Discussion

Based on data from the Chinese Nutrition and Health Surveillance (2015–2017), this analysis provides up-to-date information on the relationship between RC and MetS in Chinese adults. This study found that the average RC level in the population was 0.48 mmol/L, with males having a significantly higher level than females. The prevalence of MetS varied significantly among RC groups, with the highest prevalence found in the highest RC group. The OR of MetS increased with each quartile increase in RC. Additionally, there was a positive correlation between RC and each component of MetS, with the prevalence of each component increasing with each quartile increase in RC. RC had moderate diagnostic accuracy in identifying MetS.

Previous studies have shown significant differences in RC levels among countries or regions. For example, the median RC levels were 0.6 mmol/L in the United States, 0.53 mmol/L in southern Iran, and 0.4 mmol/L in Copenhagen [21,22,23]. In the current study, the average RC level was 0.48 mmol/L. RC was significantly correlated with age, smoking, drinking and other unhealthy lifestyles. This finding was consistent with the results of a study by Huh, which also showed higher RC levels in older individuals [24]. Additionally, our study found that as the RC quartile increased, the OR of MetS also increased, which was consistent with previous findings [25]. However, the OR of MetS in the high RC group differed significantly, possibly due to the differences in the data source. In this study, we used a nationally representative database, but variations in race, gender, and culture may have contributed to a lower prevalence of MetS [25]. 

In correlation analysis between RC and MetS components, it was found that RC had the highest correlation with Elevated TG, which is consistent with previous studies [26]. This was because RC is mainly carried by Triglyceride-rich lipoproteins, resulting in a positive correlation with TG [27]. Furthermore, the association analysis showed that RC was highly associated with Elevated TG and Low HDL-C, indicating a link between dyslipidemia and RC. Previous studies had demonstrated that the TG/HDL-C ratios were a convenient tool for detecting Insulin resistance (IR), a key component of MetS and that the risk of MetS increased with an increase in this ratio [28]. In the Q4 group, it was found that the prevalence of High BP was 60.3%, which was significantly higher than the global average. This was in line with other studies that have shown that RC elevation was an independent risk factor for hypertensive events, with a higher risk of morbidity compared to traditional risk factors [29,30]. SHI et al. found similar conclusions and found that RC increased before the development of hypertension [12]. In Model 3, it was found that individuals in the Q4 group had an OR of 2.05 for Elevated FBG. Previous studies have shown that the 13-year cumulative incidence of diabetes in the RC quartile was 8.62%, 2.49%, 12.78%, and 17.91%, respectively, and logistic regression showed that for every1 mmol/L increase in RC, the risk of new-onset diabetes mellitus increased by 144% [31]. In patients with type 2 diabetes, it has been observed that the storage of VLDL triglyceride in skeletal muscle increases, leading to an increase in RC [32]. Additionally, RC was found to be positively correlated with High WC, which was consistent with the findings of Gupta [13]. Gupta’s study showed a significant association between abdominal obesity and RC in each cohort when compared to systemic obesity.

This study was the first to investigate the relationship between RC and the number of MetS components. The results showed that as RC increased, the proportion of individuals with 2, 3, 4, and 5 MetS components also increased. 

In this study, we found that RC could be used to identify MetS, with an AUC of 0.71 and a best threshold of 0.52 mmol/L. Insulin resistance is one of the underlying causes of MetS, and commonly used evaluation indexes, such as the triglyceride-glucose (TyG) index and HOMA-IR, were used to diagnose MetS [33,34]. Previous studies found that the AUC for the TyG index and HOMA-IR was 0.654 and 0.556, respectively, which was lower than 0.71 in our study, indicating that RC could be a specific indicator to predict the incidence of MetS [35]. However, a study by Yang et al. showed that the accuracy of RC in identifying MetS was higher in males than in females, which was inconsistent with our findings. This may be due to the lack of model adjustment [36].

Overall, our study provides valuable insights into the relationship between RC and MetS in Chinese adults aged 20 years and older. However, our study has some limitations. First, it was a cross-sectional study, so we cannot establish causal relationships. Second, the use of a food frequency questionnaire for dietary assessment may be subject to recall bias. Lastly, while our results were statistically significant, further research is needed to verify the reliability of our conclusions, as there may be ethnic differences in different countries.

## 5. Conclusions

A positive correlation was presented between RC and MetS in Chinese adults aged 20 or older, and OR of MetS gradually increased with an increase in the RC quartile. RC had a positive association with MetS, each MetS component and the number of MetS components. The prevalence of each MetS component increased with an increase in the RC quartile. In addition, ROC analysis showed that the accuracy of RC in identifying MetS was normal, but the performance was better. The results of this study could provide new scientific evidence for the early prevention and control of MetS.

## Figures and Tables

**Figure 1 nutrients-16-03275-f001:**
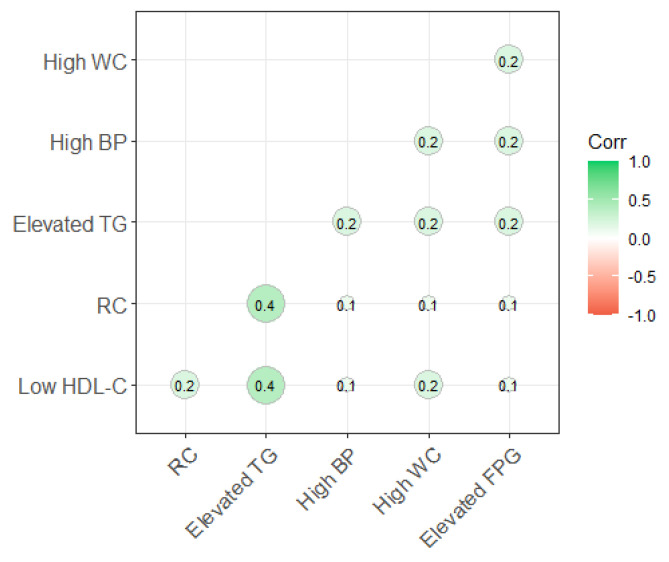
The correlation between RC and the component of MetS.

**Figure 2 nutrients-16-03275-f002:**
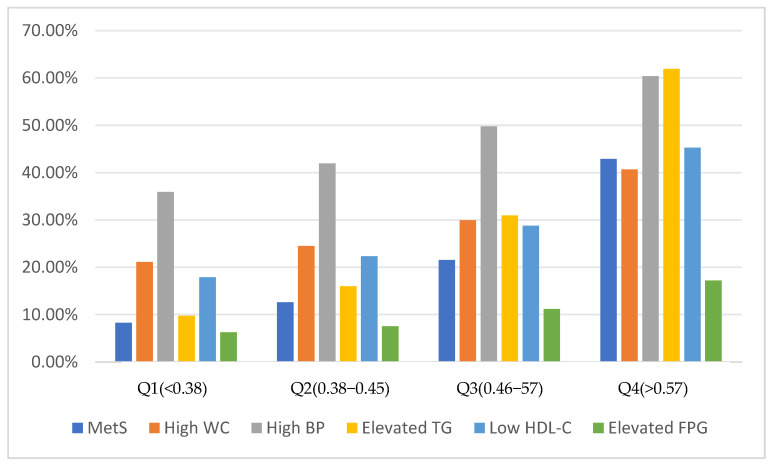
The prevalence of MetS components in different RC groups.

**Figure 3 nutrients-16-03275-f003:**
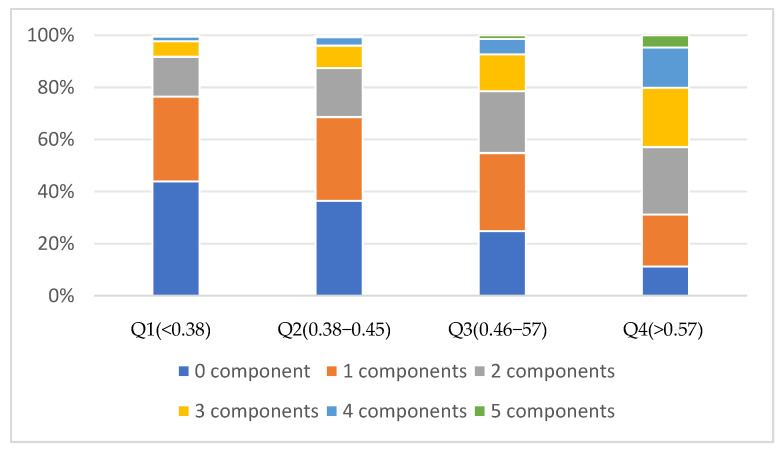
Percentage of the number of MetS components in different RC groups.

**Figure 4 nutrients-16-03275-f004:**
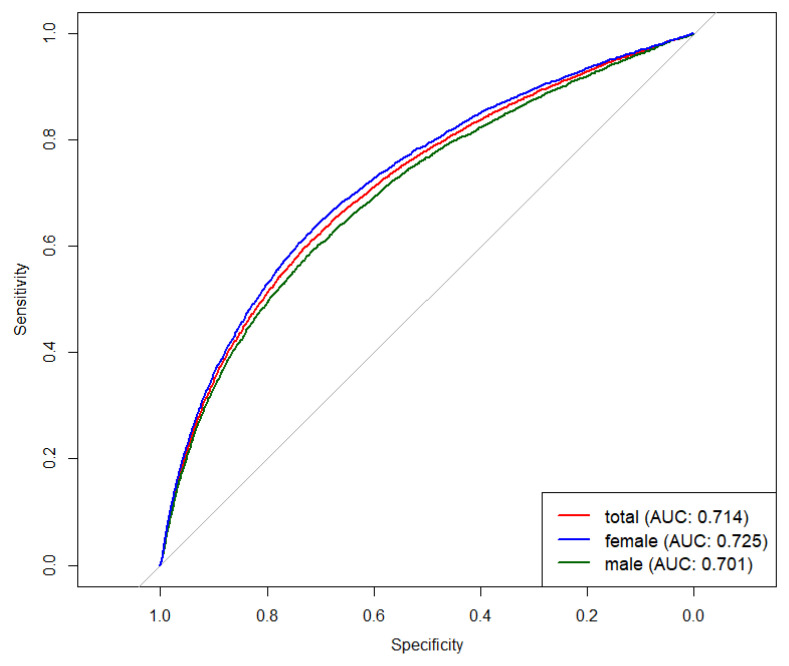
ROC analysis of RC for the identification of MetS.

**Table 1 nutrients-16-03275-t001:** Participant characteristics among adults in 2015–2017 according to sex.

	Total	Male	Female
N	65,618	30,363	35,255
Remnant Cholesterol *	0.50 ± 0.21	0.50 ± 0.21	0.49 ± 0.22
MetS, N (%) *	15,644 (23.8)	7778 (11.9)	7866 (12.0)
Age, year *	52.52 ± 14.31	53.56 ± 14.40	51.63 ± 14.16
BMI *			
Normal	33,752 (51.4)	15,741 (24.0)	18,011 (27.5)
Overweight	22,859 (34.8)	10,777 (16.4)	12,082 (18.4)
Obese	9007 (13.7)	3845 (5.9)	5162 (7.9)
Education status, N (%) *			
Primary school graduate or below	31,984 (48.7)	12,485 (19.0)	19,499 (29.7)
Middle/high school	28,681 (43.7)	15,493 (23.6)	13,188 (20.1)
College graduate or above	4953 (7.6)	2385 (3.6)	2568 (3.9)
Household yearly income per capita, N (%)			
No given	10,682 (16.3)	4861 (7.4)	5821 (8.9)
<10,000 CNY	24,473 (37.3)	11,477 (17.5)	12,996 (19.8)
10,000–20,000 CNY	16,195 (24.7)	7431 (11.3)	8764 (13.4)
>20,000 CNY	14,268 (21.7)	6594 (10.1)	7674 (11.7)
Marital status, N (%) *			
Never married	2302 (3.5)	1460 (2.2)	842 (1.3)
Married	60,425 (92.1)	28,027 (42.7)	32,398 (49.4)
Other	2891 (4.4)	876 (1.3)	2015 (3.1)
Area of the country, N (%)			
North	29,603 (45.1)	13,799 (21.0)	15,804 (24.1)
South	36,015 (54.9)	16,564 (25.2)	19,451 (29.6)
Residence location, N (%) *			
Urban	26,894 (41.0)	12,112 (18.5)	14,782 (22.5)
Rural	38,724 (59.0)	18,251 (27.8)	20,473 (31.2)
Smoking status, N (%) *			
Never smoke	44,182 (67.3)	10,215 (15.6)	33,967 (51.8)
Former smoke	4511 (6.9)	4242 (6.5)	269 (0.4)
Current smoking	16,925 (25.8)	15,906 (24.2)	1019 (1.6)
Alcohol consumption, N (%) *			
Never	40,939 (62.4)	12,696 (19.4)	28,243 (43.0)
Moderate	10,369 (15.8)	5728 (8.7)	4641 (7.1)
Excessive	14,310 (21.8)	11,939 (18.2)	2371 (3.6)
Red meat intake *			
Normal	52,093 (79.4)	22,683 (34.6)	29,410 (44.8)
Excessive	13,525 (20.61)	7680 (11.7)	5845 (8.9)
Vegetable intake *			
Normal	20,672 (31.5)	9989 (15.2)	10,683 (16.3)
Insufficient	44,946 (68.5)	20,374 (31.1)	24,572 (37.5)
Fruit intake *			
Normal	4452 (6.8)	1643 (2.5)	2809 (4.3)
Insufficient	61,166 (93.2)	28,720 (43.8)	32,446 (49.5)
Sleep time, N (%) *			
Low	13,455 (20.5)	6098 (9.3)	7357 (11.2)
Normal	44,762 (68.2)	20,838 (31.8)	23,924 (36.5)
High	7401 (11.3)	3427 (5.2)	3974 (6.1)
Physically active, N (%) *			
active	54,182 (82.6)	24,462 (37.3)	29,720 (45.3)
inactive	11,436 (17.4)	5901 (9.0)	5535 (8.4)
Laboratory results			
TC, mmol/L *	4.75 ± 0.92	4.71 ± 0.90	4.80 ± 0.93
TG, mmol/L *	1.39 ± 0.83	1.44 ± 0.87	1.35 ± 0.79
LDL-C, mmol/L *	2.96 ± 0.82	2.94 ± 0.81	2.97 ± 0.82
HDL-C, mmol/L *	1.30 ± 0.32	1.26 ± 0.33	1.33 ± 0.31
FBG, mmol/L *	5.28 ± 0.89	5.31 ± 0.91	5.26 ± 0.88
hba1c, %	4.97 ± 0.64	4.96 ± 0.63	4.97 ± 0.65

* *p* < 0.05 compared with men. Abbreviations: BMI, body mass index; TC, total cholesterol; TG, triglycerides; LDL-C, low-density lipoprotein cholesterol; HDL-C, high-density lipoprotein cholesterol; FBG, fasting blood glucose; hba1c, hemoglobin a1c.

**Table 2 nutrients-16-03275-t002:** Weighted results and characteristics among adults in 2015–2017 by RC group.

Characteristics	Total	Quartiles of Remnant Cholesterol Index				
		Q1	Q2	Q3	Q4	*p*
Remnant Cholesterol	0.48 ± 0.00	0.31 ± 0.00	0.42 ± 0.00	0.51 ± 0.00	0.75 ± 0.00	<0.01
MetS	20.3 (19.7, 20.9)	2.4 (2.1, 2.7)	3.1 (2.9, 3.3)	5.1 (4.8, 5.5)	9.7 (9.3, 10.1)	<0.01
Age, year	44.44 ± 0.13	40.53 ± 0.24	42.83 ± 0.26	46.66 ± 0.28	48.89 ± 0.27	<0.01
Sex						<0.01
Male	49.9 (49.1, 50.6)	13.4 (12.8, 14.0)	12.3 (11.7, 12.9)	12.1 (11.6, 12.7)	12.06 (11.6, 12.5)	
Female	50.1 (49.4, 50.9)	15.6 (15.0, 16.2)	12.3 (11.8, 12.8)	11.7 (11.3, 12.5)	10.55 (10.1, 11.0)	
BMI						<0.01
Normal	53.8 (53.0, 54.6)	18.3 (17.6, 19.0)	14.4 (13.8, 15.0)	12.1 (11.6, 12.6)	9.1 (8.7, 9.5)	
Overweight	32.6 (31.9, 33.3)	8.0 (7.6, 8.4)	7.4 (7.0, 7.8)	8.4 (8.0, 8.8)	8.9 (8.5, 9.3)	
Obese	13.6 (13.0, 14.1)	2.8 (2.5, 3.0)	2.8 (2.6, 3.1)	3.4 (3.0, 3.6)	4.7 (4.3, 5.0)	
Education status						<0.01
Primary school graduate or below	36.8 (36.1, 37.5)	9.3 (8.9, 9.7)	8.9 (8.5, 9.2)	9.4 (9.1, 9.8)	9.2 (8.8, 9.5)	
Middle/high school	48.1 (47.3, 48.9)	14.2 (13.7, 14.8)	11.7 (11.1, 12.2)	11.4 (10.9, 11.8)	10.9 (10.4, 11.3)	
College graduate or above	15.1 (14.4, 15.9)	5.5 (5.0, 6.0)	4.0 (3.6, 4.5)	3.0 (2.7, 3.4)	2.6 (2.3, 2.9)	
Household yearly income per capita						<0.01
No given	15.9 (15.3, 16.4)	4.8 (4.4, 5.2)	3.9 (3.6, 4.2)	3.7 (3.4, 3.9)	3.5 (3.2, 3.7)	
<10,000	35.4 (34.7, 36.1)	9.4 (9.0, 9.8)	8.7 (8.3, 9.1)	8.8 (8.4, 9.2)	8.4 (8.0, 8.8)	
10,000–20,000	24.4 (23.7, 25.1)	7.2 (6.8, 7.6)	6.1 (5.6, 6.5)	5.7 (5.4, 6.11)	5.4 (5.1, 5.8)	
>20,000	24.3 (23.7, 25.0)	7.6 (7.1, 8.1)	5.8 (5.5, 6.2)	5.6 (5.2, 6.00)	5.3 (5.0, 5.6)	
Marital status						<0.01
Never married	10.5 (9.8, 11.3)	4.3 (3.8, 4.8)	2.8 (2.5, 3.2)	2.0 (1.6, 2.4)	1.3 (1.1, 1.6)	
Married	85.9 (85.2, 86.7)	23.9 (23.3, 24.6)	21.0 (20.4, 21.6)	20.8 (20.2, 21.4)	20.2 (19.7, 20.8)	
Other	3.6 (3.4, 3.8)	0.8 (0.7, 0.9)	0.8 (0.7, 0.9)	1.0 (0.9, 1.1)	1.1 (0.9, 1.2)	
Area of the country						<0.01
North	47.3 (46.5, 48.1)	14.5 (13.9, 15.1)	11.5 (11.0, 12.0)	10.6 (10.1, 11.1)	10.7 (10.3, 11.2)	
South	52.7 (51.9, 53.5)	14.5 (13.9, 15.0)	13.1 (12.6, 13.6)	13.3 (12.8, 13.8)	11.9 (11.4, 12.3)	
Residence location						<0.01
Urban	48.5 (47.7, 49.3)	15.6 (15.0, 16.3)	11.7 (11.2, 12.3)	10.8 (10.2, 11.3)	10.4 (9.9, 10.9)	
Rural	51.6 (50.7, 52.3)	13.4 (12.9, 13.8)	12.9 (12.4, 13.3)	13.1 (12.6, 13.5)	12.2 (11.8, 12.7)	
Smoking status						<0.01
Never smoke	67.8 (67.0, 68.5)	21.2 (20.6, 21.9)	16.7 (16.1, 17.3)	15.8 (15.3, 16.4)	14.0 (13.5, 14.5)	
Former smoke	5.4 (5.1, 5.7)	1.3 (1.2, 1.5)	1.2 (1.0, 1.4)	1.4 (1.2, 1.5)	1.5 (1.3, 1.7)	
Current smoking	26.9 (26.2, 27.6)	6.4 (6.1, 6.8)	6.7 (6.3, 7.1)	6.6 (6.3, 7.0)	7.1 (6.7, 7.5)	
Alcohol consumption						0.06
Never	59.2 (58.4, 59.9)	17.0 (16.4, 17.6)	14.6 (14.0, 15.1)	14.4 (13.9, 15.0)	13.2 (12.7, 13.7)	
Moderate	17.0 (16.5, 17.6)	5.3 (4.9, 5.7)	4.2 (3.8, 4.5)	3.9 (3.6, 4.2)	3.7 (3.5, 4.0)	
Excessive	23.8 (23.1, 24.5)	6.7 (6.3, 7.2)	5.8 (5.5, 6.2)	5.6 (5.2, 5.9)	5.7 (5.3, 6.1)	
Red meat intake						0.98
Normal	77.3 (76.6, 77.9)	22.5 (21.8, 23.1)	18.9 (18.3, 19.6)	18.4 (17.8, 19.0)	17.5 (17.0, 18.0)	
Excessive	22.8 (22.1, 23.4)	6.6 (6.1, 7.0)	5.6 (5.3, 6.0)	5.4 (5.1, 5.8)	5.1 (4.8, 5.5)	
Vegetable intake						0.71
Normal	30.6 (29.9, 31.3)	9.0 (8.6, 9.5)	7.6 (7.2, 8.0)	7.2 (6.8, 7.7)	6.8 (6.4, 7.1)	
Insufficient	69.4 (68.7, 70.1)	20.0 (19.3, 20.7)	17.0 (16.4, 17.6)	16.6 (16.0, 17.1)	15.8 (15.3, 16.4)	
Fruit intake						<0.05
Normal	8.2 (7.7, 8.6)	2.6 (2.3, 2.8)	2.1 (1.8, 2.4)	1.9 (1.7, 2.1)	1.6 (1.5, 1.8)	
Insufficient	91.8 (91.4, 92.3)	26.4 (25.7, 27.1)	22.5 (21.8, 23.2)	21.9 (21.3, 22.6)	21.0 (20.4, 21.6)	
Sleep time						<0.01
Low	16.5 (16.0, 17.0)	4.0 (3.8, 4.3)	4.0 (3.7, 4.3)	4.3 (4.1, 4.6)	4.1 (3.9, 4.3)	
Normal	72.2 (71.5, 72.8)	21.9 (21.2, 22.6)	17.8 (17.2, 18.4)	16.8 (16.2, 17.4)	15.7 (15.2, 16.3)	
High	11.4 (10.9, 11.9)	3.1 (2.8, 3.4)	2.8 (2.5, 3.0)	2.7 (2.9, 3.0)	2.8 (2.5, 3.0)	
Physically active						0.92
active	80.7 (80.1, 81.4)	23.4 (22.7, 24.1)	19.9 (19.3, 20.5)	19.2 (18.6, 19.8)	18.2 (17.6, 18.7)	
inactive	19.3 (18.6, 19.9)	5.6 (5.2, 6.0)	4.7 (4.3, 5.0)	4.6 (4.2, 5.0)	4.4 (4.1, 4.8)	
Laboratory results						
TC, mmol/L	4.60 ± 0.01	4.24 ± 0.01	4.44 ± 0.01	4.73 ± 0.01	5.12 ± 0.01	<0.01
TG, mmol/L	1.37 ± 0.01	0.96 ± 0.01	1.12 ± 0.01	1.38 ± 0.01	2.14 ± 0.02	<0.01
LDL-C, mmol/L	2.86 ± 0.01	2.58 ± 0.01	2.72 ± 0.01	2.97 ± 0.01	3.23 ± 0.01	<0.01
HDL-C, mmol/L	1.26 ± 0.00	1.35 ± 0.00	1.30 ± 0.00	1.25 ± 0.00	1.14 ± 0.00	<0.01
FBG, mmol/L	5.17 ± 0.01	5.01 ± 0.01	5.08 ± 0.01	5.24 ± 0.01	5.42 ± 0.01	<0.01
hba1c,%	4.88 ± 0.00	4.82 ± 0.01	4.85 ± 0.01	4.91 ± 0.01	4.95 ± 0.01	<0.01

**Table 3 nutrients-16-03275-t003:** Logistic regression analyses for the association between RC and MetS in different models.

Indicators	Group of Quartile	MetS OR (95% CI)		
		Model 1	Model 2	Model 3
RC	Q1	reference	reference	reference
	Q2	1.60 (1.38, 1.85)	1.50 (1.29, 1.73)	1.47 (1.26, 1.72)
	Q3	3.03 (2.65, 3.48)	2.64 (2.30, 3.03)	2.54 (2.19, 2.94)
	Q4	8.32 (7.30, 9.47)	7.06 (6.20, 8.03)	6.66 (5.82, 7.62)

Model 1: Variables are not adjusted; Model 2: Adjusting for sex and age on the basis of Model 1; Model 3: BMI, education status, household yearly income per capita, marital status, area of the country, residence location, smoking status, alcohol consumption, vegetable intake, fruit intake, sleep time, physically active, were adjusted on the basis of Model 2.

**Table 4 nutrients-16-03275-t004:** Logistic regression analyses for the association between RC and the components of MetS in different models.

Indicators	Group of Quartile	MetS OR (95% CI)		
		Model 1	Model 2	Model 3
High WC	Q1	reference	reference	reference
	Q2	1.21 (1.09, 1.34)	1.17 (1.06, 1.30)	1.06 (0.93, 1.21)
	Q3	1.60 (1.45, 1.76)	1.47 (1.33, 1.62)	1.16 (1.03, 1.32)
	Q4	2.56 (2.33, 2.81)	2.29 (2.08, 2.52)	1.47 (1.31, 1.66)
	*p*-values	<0.01	<0.01	<0.01
High BP	Q1	reference	reference	reference
	Q2	1.29 (1.18, 1.41)	1.15 (1.04, 1.27)	1.10 (1.00, 1.22)
	Q3	1.77 (1.62, 1.93)	1.31 (1.19, 1.44)	1.18 (1.07, 1.31)
	Q4	2.72 (2.49, 2.97)	1.86 (1.69, 2.05)	1.51 (1.37, 1.68)
	*p*-values	<0.01	<0.01	<0.01
Elevated TG	Q1	reference	reference	reference
	Q2	1.76 (1.52, 2.03)	1.71 (1.48, 1.98)	1.72 (1.49, 1.98)
	Q3	4.14 (3.62, 4.73)	3.98 (3.49, 4.54)	3.96 (3.47, 4.53)
	Q4	14.99 (13.18, 17.06)	14.31 (12.61, 16.26)	13.94 (12.31, 15.79)
	*p*-values	<0.01	<0.01	<0.01
Low HDL-C	Q1	reference	reference	reference
	Q2	1.32 (1.17, 1.48)	1.31 (1.16, 1.47)	1.31 (1.16, 1.48)
	Q3	1.86 (1.66, 2.08)	1.87 (1.67, 2.10)	1.84 (1.63, 2.06)
	Q4	3.80 (3.43, 4.22)	3.86 (3.48, 4.29)	3.57 (3.21, 3.98)
	*p*-values	<0.01	<0.01	<0.01
Elevated FPG	Q1	reference	reference	reference
	Q2	1.23 (1.07, 1.41)	1.12 (0.97, 1.28)	1.10 (0.96, 1.26)
	Q3	1.89 (1.65, 2.18)	1.51 (1.31, 1.76)	1.43 (1.23, 1.65)
	Q4	3.12 (2.74, 3.55)	2.36 (2.06, 2.70)	2.05 (1.79, 2.35)
	*p*-values	<0.01	<0.01	<0.01

Model 1: Variables are not adjusted; Model 2: Adjusting for sex and age on the basis of Model 1; Model 3: BMI, education status, household yearly income per capita, marital status, area of the country, residence location, smoking status, alcohol consumption, vegetable intake, fruit intake, sleep time, physically active, were adjusted on the basis of Model 2. Abbreviations: WC, waist circumference; BP, blood pressure; FPG, fasting blood glucose.

## Data Availability

According to the policy of the National Institute of Nutrition and Health of the Chinese Center for Disease Control and Prevention, data related to this study may not be disclosed.

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
