# Peer review of "Association between Remnant Cholesterol and Metabolic Syndrome among Chinese Adults: Chinese Nutrition and Health Surveillance (2015–2017)"

_nutrients, 2024, doi:10.3390/nu16193275_

Round 1

Reviewer 1 Report

Comments and Suggestions for Authors

The authors examined associations between remnant cholesterol and MetS as well MetS components in a large cohort aged 20 years and more.

The authors used descriptive statistics, correlation analyses, logistic regression and ROC curve analyses.

The study is well-designed and conduced, however, data presentation and language are very poor.

All section of the manuscript need detailed language improvement as well as more precise and accurate description of the data presented in tables and figures.

Abstract should be shortened.

Due to poor English several passages of the Discussion section are difficult to understand.

The authors mixed up the terms quartiles and quintiles throughout the text – please correct where appropriate.

The authors should indicate in the Statistical Analyses section how associations shown in Fig. 1 were calculated – Spearman or Pearson correlation and please indicate in the Fig. 1 legend what the values in Fig. 1 represent (correlation coefficients?).

Lanes 146-149: There is twice ‘’red meet intake’’ instead of vegetable and fruit intake – please correct.

Table 4: Please provide p-values to show whether the calculated associations were statistically significant and whether adjustment affects in addition to ORs also p-values.

Fig. 1. Please expand the legend (see above) and explain abbreviations.

Fig. 2. Please label y axis (%)  

Comments on the Quality of English Language

Very poor English.

Author Response

Comments1:

The authors examined associations between remnant cholesterol and MetS as well MetS components in a large cohort aged 20 years and more.

The authors used descriptive statistics, correlation analyses, logistic regression and ROC curve analyses.

The study is well-designed and conduced, however, data presentation and language are very poor.

All section of the manuscript need detailed language improvement as well as more precise and accurate description of the data presented in tables and figures.

Abstract should be shortened.

Due to poor English several passages of the Discussion section are difficult to understand.

The authors mixed up the terms quartiles and quintiles throughout the text – please correct where appropriate.

The authors should indicate in the Statistical Analyses section how associations shown in Fig. 1 were calculated – Spearman or Pearson correlation and please indicate in the Fig. 1 legend what the values in Fig. 1 represent (correlation coefficients?).

Lanes 146-149: There is twice ‘’red meet intake’’ instead of vegetable and fruit intake – please correct.

Table 4: Please provide p-values to show whether the calculated associations were statistically significant and whether adjustment affects in addition to ORs also p-values.

Fig. 1. Please expand the legend (see above) and explain abbreviations.

Fig. 2. Please label y axis (%)

Response 1:

I have revised the article and replaced "quintiles" with "quartiles". The explanation of Fig.1 can be found in the methodology section. I have also updated  the Covariates section and included the ORs and p-values in Table 4. Additionally, I have addressed any inconsistencies in the statistical chart.

Reviewer 2 Report

Comments and Suggestions for Authors

Interesting study that provides many descriptive and analytical epidemiological data
of metabolic variables  on the Chinese reality, with the limitations of being a
cross-sectional study. Below are some comments: 1) The English is lacking in some
points; 2) The laboratory techniques used should be specified ; 3) Please, report the
validation of the FFQ at least for used food items; 4) Clarify the relationship of the
covariates with the main relationship (cholesterol remnants and metabolic syndrome).
Are they confounders, intermediates or colliders? ;
5) In table 1 the significance tests
are quite useless due to the large sample size and the descriptive meaning of the table;
6) In table 2 , fruit intake 210
à 21.0 ; 7) In the discussion section : p 12, line 318,
this is a cross-sectional study, it finds associations, to predict maybe it is better to use
a longitudinal study.

Author Response

comments2:Interesting study that provides many descriptive and analytical epidemiological data
of metabolic variables on the Chinese reality, with the limitations of being a
cross-sectional study. Below are some comments: 1) The English is lacking in some
points; 2) The laboratory techniques used should be specified ; 3) Please, report the
validation of the FFQ at least for used food items; 4) Clarify the relationship of the
covariates with the main relationship (cholesterol remnants and metabolic syndrome).
Are they confounders, intermediates or colliders? ; 5) In table 1 the significance tests
are quite useless due to the large sample size and the descriptive meaning of the table;
6) In table 2 , fruit intake 210à 21.0 ; 7) In the discussion section : p 12, line 318,
this is a cross-sectional study, it finds associations, to predict maybe it is better to use
a longitudinal study.

response2:

The article has been revised and in the methods section, I have provided a more detailed description of the various techniques used in the laboratory. The relationship between the covariate and main variable is complex and requires further investigation. The value of 210 in Table 2 has been corrected to 21.0. In line 318, I deleted some of the unreasonable descriptions.

Round 2

Reviewer 1 Report

Comments and Suggestions for Authors

The authors improved the manuscript.

lane 175: Please replace Spearma with Spearman.

Comments on the Quality of English Language

There are still several gramatic and syntax errors throughout the text.

Author Response

comments1:

The authors improved the manuscript.

lane 175: Please replace Spearma with Spearman.

There are still several gramatic and syntax errors throughout the text.

Response1:

I have polished it through a professional English polishing website and corrected the grammar errors in this article. Incorrect spelling has also been corrected。